# Utilization of community engagement in social innovation health projects in low-and-middle income countries: A global sequential mixed methods analysis

Emmanuel Ahumuza[1]*, Patricia M. Waterous[2], Joseph D. Tucker[3], Nicholus Nanyeenya[1], Phyllis Awor[1]

**1** Department of Community Health and Behavioral Sciences, Makerere University, School of Public Health, Kampala, Uganda, **2** Department of Clinical Research, London School of Hygiene and Tropical Medicine, London, United Kingdom, **3** Institute of Global Health and Infectious Diseases, University of North Carolina, Chapel Hill, United States of America

* eahumuza2@gmail.com

## Abstract

Social innovation in health provides solutions to address healthcare delivery gaps using community engagement as a strategy. This study aimed to assess and understand the extent to which social innovation in health projects in LMICs use community engagement approaches in their processes, and how these shape and influence the outcomes of these health projects. We used a sequential mixed methods approach. Semi-structured interviews with social innovators were followed by an online survey among social innovation researchers and implementers. Interviews were recorded, transcribed and analyzed using NVivo 11, and themes informed development of the survey tool. The survey data were analyzed using STATA version-14 and descriptive statistics were presented. A total of 27 social innovators participated in semi-structured interviews, with 66.7% from Africa. The survey respondents had a mean age of 38.6 years (SD ± 11.5) and most of them were males 141 (51.5%). Community engagement in social innovation health projects was mostly utilized during problem identification 167 (62.1%), intervention identification and design 179 (66.5%), and delivery of interventions 213 (79.2%). Half of the social innovations 135 (52.1%) had very high level of community engagement, described as collaborative or shared leadership. There was differential participation of community stakeholders at different stages of the social innovation projects. Community engagement resulted into intervention acceptance, community ownership, and improved sustainability of interventions. Key sustainability strategies employed in social innovation health projects included capacity building, integration with existing health systems, partnerships, and financial interventions. More than half of the social innovation health projects used community engagement strategies across the various stages of their work. However, there is still need to create strong governance structures, co-create interventions with

**Data availability statement:** Due to ethical considerations, the data cannot be shared publicly. De-identified data can be made available upon reasonable request from the Makerere University School of Public Health Research Ethics Committee (SPHREC), which reviewed and approved the study. Requests for access may be sent to SPHREC at sphrecadmin@musph.ac.ug.

**Funding:** This work was supported by the TDR, the Special Programme for Research and Training in Tropical Diseases (grant number AP21-00342 to PA). TDR is co-sponsored by UNICEF, UNDP, the World Bank, and WHO, and receives additional funding from Sida, the Swedish International Development Cooperation Agency, to support the Social Innovation in Health Initiative. The funders had no role in study design, data collection and analysis, decision to publish, or preparation of the manuscript.

**Competing interests:** The authors have declared that no competing interests exist.

communities, and have shared responsibilities with the communities in order to attain the desirable high level of community engagement and substantiality in social innovation health projects in LMICs.

## Background

Health systems in many low-and-middle-income countries (LMICs) including Uganda, continue to face persistent challenges, including limited resources, inequitable access to care, and gaps between health services and the communities they are meant to serve. Health interventions designed at the top leadership level without the community engagement often struggle to respond effectively to the different health challenges. Hence, social innovation in health is gaining increasing attention as a complementary approach to strengthening health systems because it involves a community engagement approach [1].

The World Health Organization (WHO), through the Special Programme for Research and Training in Tropical Diseases (TDR), defines social innovation in health as community-driven solutions that are locally developed, affordable, and designed to improve access, quality, and equity in health care [2,3]. Social innovations may include new service delivery models, financing mechanisms, digital tools, or community-led initiatives that respond to real-world constraints. Importantly, WHO/TDR emphasizes that social innovation is not only about novelty, but about practical solutions that work within specific social, cultural, and health system contexts [4,5]. Social innovations in health have demonstrated the potential to address healthcare service delivery gaps in low- and middle-income countries (LMICs). For example, social innovations have increased uptake of diagnostics in Malawi, China and Peru [6] and enhanced housing conditions and mitigated environmental risks, resulting in decreased infestation rates of Chagas disease in Guatemala [7]. They have also broadened the scope of private sector pharmacy services for effective management of childhood illnesses [2] and improved access to reproductive health and maternal delivery services in Uganda [8]

While social innovations hold promise, their success is not guaranteed by design alone. Evidence from WHO/TDR-supported initiatives highlights that many innovations fail to achieve sustained impact or scale-up when they do not adequately reflect community priorities or gain local trust [9,10]. Hence, this has led to growing recognition that community engagement is a critical component of effective social innovation in health.

Community engagement refers to the active involvement of individuals, families, and community groups in identifying health problems, shaping solutions, implementing interventions, and evaluating outcomes [11,12]. Community engagement is seen as both an ethical principle and a practical strategy that enhances relevance, acceptability, and accountability in health interventions. When communities are meaningfully engaged, they contribute local knowledge, lived experience, and social networks that are often invisible to formal health systems but essential for innovation success

[13,14]. Community engagement can be used to tailor social innovations, accelerate implementation research, facilitate pragmatic clinical trials, inform local health policy, and help sustain social innovation over time [15]. The Social Innovation For Research Checklist (SIFR) recommends that all social innovation research describe the extent of community engagement [16]. In addition, the TDR/SESH/SIHI Monitoring and Evaluation Guide recommends that community engagement be a core component of assessing social innovation [9].

The relationship between social innovation and community engagement is bidirectional and mutually reinforcing. On one hand, community engagement serves as a key input into social innovation by helping innovators understand local needs, barriers, and opportunities. This participatory process increases the likelihood that innovations are culturally appropriate, trusted, and used as intended [17,18]. On the other hand, social innovations can strengthen community engagement by creating new spaces for participation, building community capacity, and fostering a sense of ownership over health solutions [19–21].

Despite this relationship between social innovation and community engagement, there have been only a few studies examining community engagement in the context of social innovation in health [22]. Previous studies of community engagement related to social innovation in health have not examined important contextual issues [23]. Some social innovation studies have focused on high-income countries [2], neglecting the many low and middle-income countries that lead and evaluate social innovation. Understanding community engagement in the context of social innovation is essential for creating sustainable, culturally appropriate and replicable solutions that have a positive impact on the communities they serve [24–26].

Therefore, the purpose of this study was to assess and understand the extent to which social innovation in health projects in LMICs use community engagement approaches in their processes, and how these shape and influence the outcomes of these health projects.

## Methods

### Study design

We used a sequential exploratory mixed methods approach [27]. The study had two sequential components, including a qualitative design with a series of semi-structured interviews with social innovators, and this was followed by a cross-sectional online survey among social innovation researchers and implementers. The qualitative design aimed to explore the experiences of social innovation researchers and implementers about the use of community engagement in their social innovation projects and the facilitators and barriers, and this would also partly inform the quantitative survey. The collected qualitative data was analyzed and some of the different themes obtained inductively informed the selection of the items for some sections in the subsequent quantitative survey. We chose a mixed methods approach because it allows for a more comprehensive understanding of community engagement in social innovations in health.

### Semi-structured interviews

We conducted 27 key informant interviews (KIIs) with social innovators who had been identified through the Social Innovation in Health Initiative (SIHI) Network [23], until data saturation was achieved [28]. SIHI organized a global compendium of 40 social innovation case studies. Each of these cases met pre-specified criteria and were vetted by two independent external individuals [9]. A brief descriptions of each of these social innovation case studies is available in the supplementary materials (S1 File). We contacted each of the social innovators identified in the 2022 version of the case compendium via email up to a total of three times. For individuals who did not respond, we found contacts at the host institution. When making an initial inquiry, we asked to interview people with knowledge about the origins and development of that specific social innovation health project over time, and we used purposive stratified sampling to ensure variability of the study participants. All interviews were conducted online via Zoom, and Teams between 7th December 2021–12th February 2022.

Each interview lasted approximately 30–45 minutes and was conducted by EA and PW who have experience in collecting qualitative data and have attained masters and PhD degrees respectively. The semi-structured interview guide was iteratively developed with feedback from social innovators in addition to personnel at the SIHI research hubs in Uganda and China.

All interviews were audio recorded and transcribed. Key elements of community engagement were extracted from the interviews using deductive and inductive coding. To enhance rigor of the analysis, a coding framework was developed by two authors, EA and PW who were involved in the data collection process. Themes were identified by exploring the links between the codes and clustering them. Codes were discussed amongst the authors until consensus was reached. Subsequently, basic themes were identified from the coded segments of data and refined. These basic themes were then organized into organizing themes and main themes that aligned with the research questions (Table 1). NVivo 11 was used for thematic analysis.

The qualitative findings revealed three levels of community engagement in social innovation in health: "shared leadership", "collaborate", and "involve". This study explored the relationship between these levels of community engagement and the sustainability strategies employed by social innovations in health.

### Online survey

Building on themes identified from the semi-structured interviews, we developed an online survey instrument. This was developed based on best practices for online surveys during COVID-19 [29]. We used a convenience sampling approach. The survey focused on social innovators, government officials, researchers, and other stakeholders who have experience with social innovation in health. Recruitment was conducted entirely online, facilitated by SIHI research hubs which shared the survey link with the target groups. The survey included domains on socio-demographic characteristics, the extent to which community engagement was being done, facilitators and barriers of community engagement, and related topics. Using the community engagement framework developed by the US Centers for Disease Control and Prevention, respondents categorized the level of community engagement in their social innovation projects as "inform", "consult", "involve", "collaborate" or "shared leadership". The survey was field tested among seven individuals at the SIHI China and Uganda hubs prior to launch. Pretesting was done to test for clarity and acceptability of the questions and the ability of the questionnaire to generate required data. Based on the feedback received, the questionnaire was improved. The online survey was conducted from 6th May 2022–27th June 2022.

Survey data were entered into MS Excel and analyzed using STATA version 14. Descriptive analysis was used to summarize quantitative data. Frequencies and percentages pertaining to levels, stages, functions, and risks and benefits of community engagement in social innovations were computed.

### Ethical considerations

The mixed methods study was approved by Makerere University School of Public Health Higher Degrees, Research and Ethics Committee (HDREC) (reference: SPH-2020–9). Written informed consent was obtained from the respondents participating in the qualitative interviews. Participation was voluntary and data of the respondents remained anonymous through data collection, analysis and dissemination. Respondents in the online survey were informed about the study's purpose, voluntary participation, confidentiality, and survey duration in the introductory section. Proceeding with the survey implied consent.

### Results

A total of 274 individuals participated in the survey. Fifty-seven percent (56.6%) of the respondents were from Africa, a quarter of them from Asia, and 16% were from Latin America and the Caribbean. The mean age was 38.6 (SD ± 11.5) (Table 2). Of the 27 key informants involved in the study, 16 were males and 18 were from Africa (S1 Table).

**Table 1. Coding framework.**

| Main Theme | Organizing theme | Basic theme |
|---|---|---|
| Stages of community engagement in social innovation health project | Determining the community health needs | • Engaging community members through focus group discussions and community dialogues to gather insights on public health needs.<br>• Utilizing feedback surveys to understand community health needs.<br>• Involvement of local leaders to better understand community health needs. |
| | Design of intervention | • Working with existing community stakeholders to set shared goals and responsibilities.<br>• Utilizing feedback and crowdsourcing to design programs |
| | Implementation/delivery of intervention | • Engaging local community stakeholders in mobilizations and awareness campaigns.<br>• Engaging local community stakeholders in promoting utilization of health services and products.<br>• Strengthening the capacity of local community stakeholders in delivering health services to the community. |
| Level of community engagement in social innovation health projects | Shared leadership | • Strong or long-term partnership structure is formed. |
| | Collaborate | • The social innovation project forms partnerships with community on each aspect of the project. |
| | Involve | • The social innovation project involves multi-participation on issues with community. |
| Sustainability strategies of social innovation health projects | Financial sustainability | • Introduction of community health insurance to reduce financial burdens on patients and donor dependency<br>• Negotiation with Government for health workers' salaries<br>• Enhancing business management of health facilities to promote self-sustainability.<br>• Diversification of funding sources through research grants, health organizations, local governments & additional revenue streams<br>• Developing cost-sharing agreements between the social innovations and the government<br>• Charging minimal user fees in private not for profit health facilities |
| | Capacity Building and Empowerment | • Empowering beneficiaries through skill development<br>• Capacity building of health workers in healthcare management<br>• Training and support for parent Groups of children with disabilities |
| | Community Ownership | • Leveraging existing community health structures<br>• Engaging local leaders in project activities and decision-making processes<br>• Co-creation of health projects/interventions with local community stakeholders<br>• Ongoing community engagement and dialogue |
| | Integration with Existing Health Systems | • Incorporating specific health interventions into existing health systems<br>• Integrating social innovations into policy frameworks<br>• Developing transition plans for incorporating the program into public health systems early<br>• Integration and expansion through government collaboration |
| | Partnerships and Collaboration | • Collaboration with diverse external stakeholders to leverage expertise and resources<br>• Strategic partnerships for capacity building |
| Risks and benefits of community engagement in social innovation health projects | Benefits of Community Engagement | • Improves community ownership<br>• Enhances sustainability and continuity of services<br>• Increases health service utilization<br>• Helps to design the intervention that address the actual needs of the community<br>• Enhances trust between the communities and health programs |
| | Risks of community engagement | • Time-consuming<br>• Costly process<br>• Risk of rejection |

**Global Public Health**
PLOS

**Table 2. Socio-demographic characteristics of respondents involved with social innovation, 2022 (N = 274).**

| Variable | Frequency, n = 274 (%) |
| --- | --- |
| **Continent** | |
| Africa | 155 (56.6) |
| Asia | 71 (25.9) |
| LAC[1] | 44 (16.0) |
| North America | 2 (0.7) |
| South America | 1 (0.4) |
| Europe | 1 (0.4) |
| **Sex at Birth** | |
| Male | 141 (51.5) |
| Female | 133 (48.5) |
| **Age** [Mean (SD)] | 38.6 (SD ± 11.5) |
| **Highest education completed** | |
| Secondary school | 3 (1.1) |
| Certificate (tertiary) | 2 (0.7) |
| Diploma | 3 (1.1) |
| Bachelor's degree | 82 (29.9) |
| Master's degree | 138 (50.4) |
| PhD | 46 (16.8) |
| **Type of organization** | |
| Government | 59 (21.5) |
| University | 97 (35.4) |
| NGO[2] | 63 (23.0) |
| Private sector | 34 (12.4) |
| Others | 21 (7.7) |
| **Occupation** | |
| Researcher | 148 (54.0) |
| Health professional | 34 (12.4) |
| Academia | 25 (9.1) |
| Public health professional | 18 (6.6) |
| Health program manager | 13 (4.7) |
| Social worker | 13 (4.7) |
| Others | 23 (8.4) |

Type of organization: Others - Freelance, Research centre, Health College, Charitable organization, Currently not working; Occupation - Health program directors, Administrators, Business/entrepreneurs, Communications professionals, Civil servants, Finance officers, Grants manager, Politician, and Software Engineer.

[1] Latin America and the Caribbean

[2] Non-governmental organization

## Stages and activities of community engagement in social innovation health projects

Survey findings (Table 3) found that most of the social innovation health projects engaged communities during delivery of interventions 213 (79.2%), intervention identification and decision making (design) 179 (66.5%), and problem identification 167 (62.1%). A total of 167 (62.1%) innovations used qualitative research studies, 149 (55.4%) advocacy of health need in

**Table 3. Stage of the social innovation at which community engagement was utilized, and the community engagement activities utilized.**

| Variable | Frequency, n (%) |
|---|---|
| **Stage of the social innovation at which community engagement was utilized (n = 269)** | |
| During implementation/delivery of the intervention | 213 (79.2) |
| During intervention identification and decision making | 179 (66.5) |
| During problem identification | 167 (62.1) |
| During evaluation of the solution | 155 (57.6) |
| During scaling the intervention/Solution | 121 (45.0) |
| In managing the resources | 83 (30.9) |
| **The community engagement activities that were utilized in the social innovation (n = 269)** | |
| Qualitative research study | 167 (62.1) |
| Advocacy of health need in community | 149 (55.4) |
| Stakeholder consultation | 146 (54.3) |
| Quantitative research study | 126 (46.8) |
| Co-creation workshop or similar participatory event | 118 (43.9) |
| Community advisory board | 92 (34.2) |
| Crowdsourcing open call | 52 (19.4) |
| Crowdsourcing designation | 25 (9.3) |
| Monitoring and Evaluation | 131 (48.7) |
| **Conducted an assessment for community engagement in the social innovations (n = 269)** | |
| Feedback from community members on the activity | 184 (68.4) |
| Number of respondents at community engagement meetings | 156 (58.0) |
| Qualitative research interviews | 137 (50.9) |
| An evaluation of the community engagement activity | 122 (45.4) |
| Number of workshops/community engagement meetings | 120 (44.6) |
| Social media analytics of shares, likes and re-tweets | 46 (17.1) |

community and 146 (54.3%) used stakeholder consultations to engage communities. Two-thirds of respondents assessed community engagement through getting feedback from community members on the activities 184 (68.4%).

Similarly, when discussed with key informants, qualitative findings revealed that more than three-quarters of social innovations (23/27) engaged communities during delivery of interventions. Local community stakeholders were involved in community mobilization, creating awareness and promoting utilization of health services and products. Social innovations also increased the capacity of the community stakeholders (e.g., health care workers, community health workers, teachers and community groups) to provide health services to the community.

*"We were working with county health teams and the Ministry of Health to implement a community health worker program and so basically recruiting and training community members to provide a diagnostic preventative and curative services to the communities".* (KI 13, Liberia)

About one half of social innovations (12/27) involved communities to determine their health needs. Community members engaged in focus group discussions, community dialogues, and feedback surveys to discuss their needs. In some instances, the local leaders, district health officials and community health workers were also involved.

*"to understand the challenges the population was facing with accessing their health care was the first stage of community engagement, this was mainly through focus group discussions and feedback surveys".* (KI, 02, Uganda)

We noted that during the design stage, two-thirds of social innovations (17/27) worked with trusted existing community stakeholders such as churches, community groups, community health workers, local leaders or employers to set shared goals and responsibilities. Feedback surveys, interviews and crowdsourcing were utilized to design services appropriate to the focus population and to design and pre-test education materials and programs. More than half of the social innovations (15/27) involved communities in evaluation of the interventions through community dialogues, surveys, and qualitative and quantitative research.

### Levels of community engagement in social innovation health projects

Most of the survey respondents categorized their social innovation projects under collaborative community engagement 88 (34.0%), followed by shared leadership 47 (18.1%), community involvement 63 (24.3%), community consultation 32 (12.4%) and finally community information 29 (11.2%) levels of community engagement (Table 4).

In qualitative interviews, almost half of the social innovations (12/27) that described having "shared leadership" ascribed this to strong governance structures created within the community. Social innovations also co-created the interventions with community stakeholders and set shared goals, roles or responsibilities.

*"So in each of the programs in the communities we went to, we created a long term governance structure where we agreed together on sort of what outcomes we wanted to achieve and so who [is] responsible for doing what".* (KI, 01, South Africa)

A quarter of the social innovation health projects (7/27) had "collaborate" as the level of community engagement. Community stakeholders and members were involved at different stages of the innovation and this built trust with the

**Table 4. Levels and functions of community engagement in social innovations.**

| Variable | Frequency, n (%) |
|---|---|
| **Level of community engagement (n = 259)** | |
| Shared leadership | 47 (18.1) |
| Collaborate | 88 (34.0) |
| Involve | 63 (24.3) |
| Consult | 32 (12.4) |
| Inform | 29 (11.2) |
| **Function of community engagement (n = 269)** | |
| Empowerment (Planning and managing health activities by the community using professionals as resources and facilitators) | 186 (68.0) |
| Mobilization (So that people will eventually do what the professional advises) | 171 (63.6) |
| Design Interventions that solve community challenges | 177 (65.8) |
| Advocate and gain support/uptake for intervention/ project | 169 (62.8) |
| Collaboration (Communities contribute time, materials and/or money, but with the professional defining needs) | 167 (62.1) |
| Increase diverse voices and be inclusive | 125 (46.5) |

community. Whereas another quarter of the innovations (8/27) exhibited "involve" as the manner of community engagement. Communities were involved in different activities including health promotion and outreach programs as well as providing feedback on services received.

### Functions of community engagement in social innovation health projects

About two-thirds of the respondents reported that they utilized community engagement for empowerment 186 (68.0%), designing interventions 177 (65.8%), mobilization 171 (63.6%), advocacy and uptake of projects 169 (62.8%) as well as for collaboration 169 (62.8%) (Table 4). Two-thirds of key informants (18/27) said that social innovations empowered community representatives (e.g., community health workers, teachers, and community members). Beneficiaries were also equipped with knowledge and skills to take responsibility and control of their health.

> *"We had to train VHTs to do a breast self-examination so this knowledge was passed out to the women in the villages to do self-breast examinations on themselves".* (KI 22, Uganda).

We also noted that more than half of the social innovations engaged community members and/or community representatives such local leaders, churches, community groups, employers and district local government officials to advocate for support or uptake the interventions. During community mobilizations, community representatives provided health education and promotion services to communities though door-to-door and community dialogue meetings. We however noted that community representatives referred high risk cases identified in the communities to the health facilities.

> *"Community health workers do home visits to mothers three weeks during pregnancy and three weeks after delivery, providing messages on health promotion and disease prevention".* (KI 25, Peru)

### Benefits and risks for community engagement in social innovation health projects

Respondents acknowledged that community engagement was beneficial for community empowerment 191 (71.0%), improving utilization of services 184 (68.4%), improving interventions 183 (68.0%), sustainability 175 (65.1%) and community ownership 173 (64.1%) (Table 5). More than half of the key informants reported that engaging communities in social innovations improved community ownership through co-creation, intervention acceptance and participation in social accountability. Key informants reported that working with governments enabled patients to be absorbed into the public health system, and community health workers ensured continuity of service delivery even after end of project activities. Other key informants noted that involving the communities in social innovations improved health service utilization, sustainability of interventions, addressed myths and mistrusts, as well as empowered communities to take charge of their health needs.

> *"It's a cornerstone for sustaining health innovations in our communities because it also creates ownership of the program, it triggers innovation or feedback from key stakeholders that is critical for improving a program. Our programs have largely been formed by feedback that has come [from] engaging communities".* (KI 08, Burundi)

The common risks of community engagement in social innovations included the process being time consuming 153 (56.9%), low literacy of community members that may hinder their understanding of the interventions or their role in contributing 125 (46.5%), people derailing or changing initial ideas 114 (42.4%) and the process being expensive 112 (41.6%) (Table 5).

Similarly, qualitative findings revealed that the process of engaging the community in social innovations is time consuming and expensive. One key informant highlighted the risk of prolonged polishing and approval of interventions: *"It prolongs the process because every other time you have to go to the community, it makes the process of approval and*

**Table 5. Benefits, risks and useful resources for community engagement.**

| Variable | Frequency, n (%) |
|---|---|
| **Benefits of robust community engagement in social innovations (n = 169)** | |
| Improved community ownership | 173 (64.1) |
| Community empowerment | 191 (71.0) |
| To improve the intervention | 183 (68.0) |
| For sustainability | 175 (65.1) |
| To solicit beneficiary/end-user perspectives | 117 (43.5) |
| To reduce mistrust in activities | 133 (49.4) |
| Enhance sharing responsibilities and resources | 115 (42.8) |
| Improves utilization of health services | 184 (68.4) |
| **Risks of community engagement in social innovations (n = 269)** | |
| Expensive | 112 (41.6) |
| Takes too much time | 153 (56.9) |
| People can derail or change initial ideas | 114 (42.4) |
| Lack of expertise in the team to conduct community engagement | 87 (32.3) |
| Community can reject your ideas | 103 (38.3) |
| Could contribute to confusion or misunderstanding the intervention | 77 (28.6) |
| Low literacy of community members that they may not or do not understand the interventions or may not have a role to contribute | 125 (46.5) |

*feedback longer"* (KI 02, Uganda). We noted that social innovations could be rejected if the communities have a poor understanding of the intervention and fail to involve cultural, political/local and religious leaders.

### Community engagement and sustainability of social innovation health projects

The qualitative study explored the sustainability strategies adopted by social innovation health projects and their relationship to the three high levels of community engagement, namely "shared leadership", "collaborate", and "involve". Findings are based on five main themes that were identified, including financial sustainability; capacity building and empowerment; community ownership; integration with existing health systems; and partnerships. These certain strategies intersect across community engagement levels (Fig 1). Financial interventions for sustainability, and capacity building and empowerment strategies are crucial across the three top levels of community engagement. Under shared leadership and collaborative levels, community ownership and integration with existing health systems were also highlighted.

The financial sustainability of social innovations in health was realized through a multifaceted approach that minimized reliance on external donors and ensured continuity of health service utilization. This approach also enhanced the self-sufficiency of health facilities. The strategies employed by various projects include: community health insurance, charging minimal user fees, diversifying funding sources through partner health organizations and local governments, cost-sharing agreements between the social innovations and the government, and improving health facility management.

Integration with existing health systems is another important strategy for ensuring the sustainability of social innovations in health. This involved incorporating certain operations of the social innovations, such as disease testing services, into the public healthcare system, aligning with public health systems for regular funding, and integrating crowdsourcing into national health frameworks. Collaborating with governments for service expansion and early integration into public health systems also supports sustainability. These approaches were employed by the Comprehensive Health Approach to Fight Chagas Disease (Guatemala), Seal of Health Governance (Philippines), Social Entrepreneur to Spur Health (China),

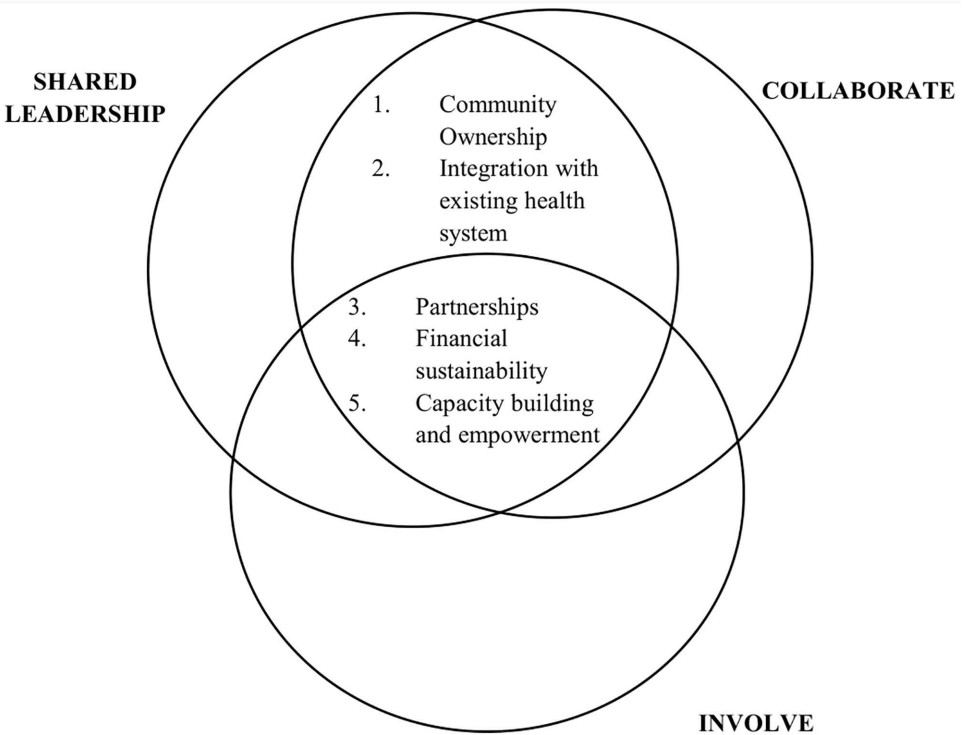

**Fig 1. Sustainability strategies employed in social innovation health projects.**

Kheth'impilo Pharmacist Assistant (South Africa), Health Child Uganda, Kyaninga Child Development Centre (Uganda), and the SMS Hub Leprosy Case Management System projects (Mozambique).

Capacity building and empowerment strategies were integral to sustaining social innovations in health. The focus was on enhancing skills and economic opportunities which enabled beneficiaries to generate income and improve their quality of life. Strengthening health facilities involved improving management practices and providing targeted training to health workers to contribute to building a resilient and capable workforce. Training and support for parent groups of children with disabilities ensured continued support for these children.

Community ownership was vital for the sustainability of social innovations in health. This strategy involved leveraging existing community health structures such as community health promoters to ensure ongoing engagement and reporting on health issues. Engaging local leaders in project activities and decision-making processes strengthened collaboration and ensured that services were iteratively refined to meet evolving community needs. Co-creating with local stakeholders ensured that projects were tailored to the community's needs.

*"You have to co-create, and if you don't co-create with the all local players then that's not sustainable… we worked with our partners to co-create how we would go to the community for example we worked very closely with the government on creating their educational materials for the patients, creating the educational programs for the private sector, GPs, and then working closely with the churches, employers in the support groups to co-create how they would run the education programs, how to reach out to patients so for me that's the only model that works ultimately."* (KI 01, South Africa)

Partnerships played a crucial role in the sustainability of social innovations in health. Working with external stakeholders, such as the government ministries and international organizations strengthened project sustainability by leveraging

existing platforms and resources. Establishing strategic partnerships supported training, management, capacity building, and policy initiatives. Lobbying for support from partner organizations ensured the provision of necessary inputs and drugs, crucial for operations like ultrasound clinics.

*"We partnered with the Management of the Port and even we also sought the support of the Local Government to help them in the trainings, management of the organization, building the capacity of the organization and even we also sought the support of the local government as a resolution, as a mode of sustainability mechanism of the project not to allow the renewal of the business permits unless they pay their premium contribution to PhilHealth that was the design of the program…We are able to engage the local government unit to pass an Ordinance to require all tricycle drivers to enroll in the (health insurance) program"* (KI 01, Philippines)

## Discussion

This mixed-methods study assessed the extent to which social innovation in health projects in LMICs use community engagement approaches in their processes. Our findings show that more than half of the social innovation health projects used community engagement strategies in their work. Communities were mostly used to identify community health needs, design interventions, and deliver interventions. There was differential participation of communities at different stages of community engagement in social innovations. The study also highlights activities of community engagement at different project stages, and functions of community engagement in social innovations such as empowerment, mobilization and advocacy for uptake of interventions.

The study found that community engagement was more common during problem identification, design and delivery of interventions compared to the management of resources and scaling up of interventions. This highlights key stages where collaboration with communities is more deliberate. The emphasis on problem identification suggests a participatory approach in understanding the specific health needs of the community, while active involvement in the design and delivery stages implies a hands-on role for communities in shaping and implementing social innovations in health. Therefore, community engagement ensures that interventions are not only locally relevant but also accepted, embraced and sustained by the community [30]. The findings are consistent with studies supporting the involvement of communities at several stages of developing a social innovation [31–33]. We noted differential participation of community members or stakeholders at different stages of social innovations [34]. For example, during problem identification, social innovations mostly engaged district local government officials, local leaders as well as health workers. At the design stage, social innovations mostly engaged community groups, religious groups, community health workers, local leaders and employers who knew communities well and were trusted by community. This implies that in order to effectively engage communities in social innovations, it is important to consider who to involve at each stage.

This study highlights the functions of community engagement in social innovation health projects, with primary focus on community empowerment, which plays an instrumental role in increased utilization of health services. Key informants noted that social innovations in health built the capacity of community health workers and teachers to offer health promotion and healthcare services within communities and schools, demonstrating the crucial role of empowered communities in driving impactful health initiatives. These findings align with prior studies suggesting that enhancing the capacity of community health workers improves health service utilization and health outcomes within communities [35,36]. Community stakeholders are trusted by the communities and therefore help to build public trust and commitment to health interventions [35,37,38]. Additionally, key informants noted that social innovations provided communities with skills and knowledge to gain more control over their decisions concerning their health and health needs.

Understanding the sustainability of health interventions and programs has received significant attention in recent years [39,40]. This focus emphasizes the need for strategies that ensure the longevity and effectiveness of health interventions. Community ownership is a critical factor for sustaining social innovation health projects. Our findings align with previous

research, highlighting the importance of engaging community stakeholders to foster a sense of ownership. This ensures continuous refinement of services based on community needs. Community ownership also enhances participation and adaptability, leading to improved acceptance and utilization of health services [41–44].

Similar to previous studies, establishing and maintaining strategic partnerships with external stakeholders increases the sustainability of health interventions [10,45]. Partnerships help in leveraging existing platforms and resources, which is crucial for sustaining health innovations. For instance, collaboration with government ministries and international organizations can provide essential support for training, management, and capacity building. Working within existing systems was also highlighted as a sustainability strategy in previous studies [10,46,47]. By integrating health interventions into broader policy initiatives and utilizing existing governmental and organizational structures, projects can enhance their reach and effectiveness.

Capacity building was identified as a fundamental strategy for sustaining social innovations in health. Previous studies show that maintaining workforce skills through ongoing training and support is essential for the sustainability of evidence-based interventions [45,48,49]. Our study corroborates this, showing that projects focus on strengthening health facilities and providing targeted training to community stakeholders. This approach not only builds a resilient and capable workforce but also enhances the overall effectiveness of the health interventions [49].

Our study had a number of strengths and limitations: The mixed methods approach, combining a survey and qualitative interviews provided a holistic understanding of community engagement in social innovation health projects. The incorporation of key informant interviews added depth to the qualitative findings, offering valuable perspectives from those directly involved in social innovation health projects. Lastly, the inclusion of a diverse sample, covering various social innovations health projects from different regions in LMICs enhanced the generalizability of findings.

The survey's reliance on self-reported data may have introduced bias, as respondents might have provided socially desirable responses. However, anonymity was assured during data collection to encourage honest responses. The study did not explore the extent of community engagement in relation to the effectiveness, and scalability of social innovations in health. Future research could delve into these aspects to offer a more detailed understanding of the dynamics between community engagement and the sustainability, governance structures and effectiveness of social innovations in health.

## Conclusion

Our study demonstrated that more than half of the social innovation health projects used community engagement strategies across the various stages of their work, mainly during intervention delivery and problem identification. However, there is still need to create strong governance structures, co-create interventions with communities, and have shared responsibilities with the communities in order to attain the desirable high level of community engagement and substantiality in social innovation health projects in LMICs.

## Supporting information

**S1 Table: Socio-demographic characteristics of the key informant participants involved in identified social innovations.**
(DOCX)

**S1 File. Description of Social Innovation in Health Case Studies.**
(DOCX)

## Acknowledgments

The authors would like to thank innovators/project leads of the following social innovations: Broadreach GP Down-Referral model; The Medical Concierge Group Limited; Health Child Uganda; Noora Health; Safe Water And Aids Project; Seal of Health Governace; Kyaninga Child development Centre; SESH China; Riders for Health; Learner Treatment Kit;

Lifenet Burundi; Kaundu Community based Insurance Malawi; Last Mile Health; Living Good Uganda; Eco-Health Health Approach to Chagas Disease, Guatemala; Kheth'impilo Pharmacist Assistant Training Programme; Schistosomiasis Control Initiative; SMS-Hub Leprosy Case Management System; Action for women and Awakening in Rural Environment (AWARE) Uganda; Bwindi Mother's waiting Hostel; Drug Shop Integrated Management of Childhood Illness; Imaging the World Africa; Everyday Family Health Plan; National Telehealth System; Mothers of the river; Centre for the Development of Scientific Research; and Comprehensive Health Approach to Fight Chagas Disease. The authors wish to thank the SIHI secretariat and all SIHI hubs for their assistance and support in contacting and scheduling interviews with social innovators in their respective countries. The authors would also like to thank the SIHI hub members and social innovation researchers for participating in the online survey on community engagement.

## Author contributions

**Conceptualization:** Emmanuel Ahumuza, Patricia M Waterous, Joseph D Tucker, Phyllis Awor.

**Formal analysis:** Emmanuel Ahumuza, Patricia M Waterous, Joseph D Tucker.

**Funding acquisition:** Patricia M Waterous, Phyllis Awor.

**Methodology:** Emmanuel Ahumuza, Patricia M Waterous, Joseph D Tucker, Nicholus Nanyeenya, Phyllis Awor.

**Validation:** Emmanuel Ahumuza, Patricia M Waterous.

**Writing – original draft:** Emmanuel Ahumuza, Patricia M Waterous, Joseph D Tucker, Nicholus Nanyeenya, Phyllis Awor.

**Writing – review & editing:** Emmanuel Ahumuza, Patricia M Waterous, Joseph D Tucker, Nicholus Nanyeenya, Phyllis Awor.

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
