## [Decision Letter · Decision Letter 0]

5 May 2025

PGPH-D-25-00549

Community Engagement in Social Innovation Health Projects: A Global Sequential Mixed Methods Analysis

Dear Dr. Ahumuza,

Thank you for submitting your manuscript to PLOS Global Public Health. After careful consideration, we feel that it has merit but does not fully meet PLOS Global Public Health’s publication criteria as it currently stands. Therefore, we invite you to submit a revised version of the manuscript that addresses the points raised during the review process.

We look forward to receiving your revised manuscript.

Kind regards,

Damen Haile Mariam, MD, MPH, PhD

Academic Editor

Additional Editor Comments (if provided):

Reviewer 1:

General -

- The relationship between community engagement (CE) and social innovation (SI) is not quite clear. The writing should provide clarity on whether CE facilitates social innovation or SE improves CE. The clarity here would determine backgrounds where problems/gaps are stated, the methods, findings and discussions.

Introduction -

- The argument "This approach [SI] prioritizes bottom-up development, empowering communities to actively shape health programs and services, thereby enhancing sustainability and accountability1. Social innovations differ from technical innovations by focusing on societal challenges through participatory solutions rather than market-oriented products or technologies". This creates a confusion where CE inherently has similar characteristics. So, which is which? what is the relationship between CE and SI? Which one contributes and benefits or what is the dual connection between the two? Further in the background section there is a statement that reads as "Robust community engagement is a foundation of social innovation. Community engagement can be used to tailor social innovations, accelerate implementation research, facilitate pragmatic clinical trials, inform local health policy, and help sustain social innovation over time". This fuels the confusion over the connection between CE and SI. The ladders of CE is widely documented to be inform, consult, involve, collaborate and empower (Harvard Catalyst. Community engagement: Innovation and improvement in public health via community engagement and research (nd). Am not sure if the ladder of CE for SI is different from these and why? Note - available evidence have established CE aims to empower the community to own and lead the change process.

Similarly, the connection between community engagement and social innovation health projects needs to be well articulated. What is the role of CE and SI in health projects and what makes such projects innovative? Is it CE or SI components of the project and consequent outcome or both?

That said, the introduction section needs to clarify existing experiences and gaps that may have called to document this which is grossly missing and in fact contributed to crude objective.

Objective -

- "The purpose of this study was to examine and understand community engagement in social innovation in health projects in LMICs". This is too general "examine and understand" what of CE and SI and in which health projects?

Methods -

- Sequential mixed method approach was used which is great. Nonetheless, except a mention of "The initial qualitative data informed the selection of items in the subsequent quantitative survey", it is not clear what elements of data were sought with the initial qualitative method to inform the survey. Neither was it clear on what were the measurements in the survey except a very general statement that reads, "...it allows for a more comprehensive understanding of community engagement in social innovations in health..... focused on social innovators, government

officials, researchers, and other stakeholders who have experience with social innovation in health... The survey included domains on socio-demographic characteristics, community engagement breadth and depth, facilitators and barriers of community engagement, and related topics". These needs to be fleshed in terms of a) why quali was done before and to generate what type of variables for survey and what the survey intended to measure b) What pre selection were used to select cases for the interviews from the global compendium of 40 social innovation case studies needs to be clearer c) specific details to such statements as: community engagement breadth and depth" including how these were measured.

Although the design gives an impression to inform selection of items for the survey, there are findings from qualitative which offers an impression that the qualitative part of this study goes beyond informing the selection of items as stated in the method section. This needs to be corrected.

Results -

- While I believe result section could be improved with clarity on some the comments above, I would love to know more about what Table 3 in the result section intends to impart? Why multiple responses? what is the difference between stage at which community engagement was utilized and community engagement utilized in SI? What does assess community engagement in social innovations mean? Check number of participants 369 against 274???

The qualitative findings need to align with the theme of the finding which should be aligned with the overarching questions (hopefully this could be clarified by further specifying the objectives.

Discussion and conclusion -

- It is important to identify some topical finding from this study and interpret each with available evidence or experiences. I am not sure if "Our findings show that half of the social innovation health projects had a very high-level engagement, categorized as either “shared leadership” or “collaborate”" While the qualitative findings and quotes do not suggest this. You may need to check your data. The statement in the conclusion section "...this study demonstrates moderate to high community engagement across various stages

of social innovation in health" doesn't seem have foundation. Neither measurements were clearer in the method section nor, the findings demonstrated this.

Reviewer 2:

The manuscript is well organized, presented, and written in Standard English. The results of the research are well discussed in the light of the available related literature and the conclusions are supported by data drawn from mixed methods analysis.

However, the title as well as the purpose of the manuscript needs to be specific in terms of geographical scope (such as LMICs as indicated in the manuscript).

Moreover, the authors need to clearly show the originality of the results of their research based on exhaustive and critical review of available related literature

Reviewers' comments:

Reviewer's Responses to Questions

**Comments to the Author**

1. Does this manuscript meet PLOS Global Public Health’s publication criteria ? Is the manuscript technically sound, and do the data support the conclusions? The manuscript must describe methodologically and ethically rigorous research with conclusions that are appropriately drawn based on the data presented.

Reviewer #1: Partly

Reviewer #2: Partly

2. Has the statistical analysis been performed appropriately and rigorously?

Reviewer #1: Yes

Reviewer #2: Yes

3. Have the authors made all data underlying the findings in their manuscript fully available (please refer to the Data Availability Statement at the start of the manuscript PDF file)?

Reviewer #1: Yes

Reviewer #2: No

4. Is the manuscript presented in an intelligible fashion and written in standard English?

Reviewer #1: Yes

Reviewer #2: Yes

5. Review Comments to the Author

Reviewer #1: I found this submission very interesting focusing on an area that is contemporary and critical for all research and development initiatives. I have several concerns that may have to be given attention.

General - The relationship between community engagement (CE) and social innovation (SI) is not quite clear. The writing should provide clarity on whether CE facilitates social innovation or SE improves CE. The clarity here would determine backgrounds where problems/gaps are stated, the methods, findings and discussions.

Introduction: The argument "This approach [SI] prioritizes bottom-up development, empowering communities to actively shape health programs and services, thereby enhancing sustainability and accountability1. Social innovations differ from technical innovations by focusing on societal challenges through participatory solutions rather

than market-oriented products or technologies". This creates a confusion where CE inherently has similar characteristics. So, which is which? what is the relationship between CE and SI? Which one contributes and benefits or what is the dual connection between the two? Further in the background section there is a statement that reads as "Robust community engagement is a foundation of social innovation. Community engagement can be used to tailor social innovations, accelerate implementation research, facilitate pragmatic clinical trials, inform local health policy, and help sustain social innovation over time". This fuels the confusion over the connection between CE and SI. The ladders of CE is widely documented to be inform, consult, involve, collaborate and empower (Harvard Catalyst. Community engagement: Innovation and improvement in public health via community engagement and research (nd). Am not sure if the ladder of CE for SI is different from these and why? Note - available evidences have established CE aims to empower the community to own and lead the change process.

Similarly, the connection between community engagement and social innovation health projects needs to be well articulated. What is the role of CE and SI in health projects and what makes such projects innovative? Is it CE or SI components of the project and consequent outcome or both?

That said, the introduction section needs to clarify existing experiences and gaps that may have called to document this which is grossly missing and in fact contributed to crude objective.

Objective: "The purpose of this study was to examine and understand community engagement in social innovation in health projects in LMICs". This is too general "examine and understand" what of CE and SI and in which health projects?

Method: Sequential mixed method approach was used which is great. Nonetheless, except a mention of "The initial qualitative data informed the selection of items in the subsequent quantitative survey", it is not clear what elements of data were sought with the initial qualitative method to inform the survey. Neither was it clear on what were the measurements in the survey except a very general statement that reads, "...it allows for a more comprehensive understanding of community engagement in social innovations in health..... focused on social innovators, government

officials, researchers, and other stakeholders who have experience with social innovation in health... The survey included domains on socio-demographic characteristics, community engagement breadth and depth, facilitators and barriers of community engagement, and related topics". These needs to be fleshed in terms of a) why quali was done before and to generate what type of variables for survey and what the survey intended to measure b) What pre selection were used to select cases for the interviews from the global compendium of 40 social innovation case studies needs to be clearer c) specific details to such statements as: community engagement breadth and depth" including how these were measured.

Although the design gives an impression to inform selection of items for the survey, there are findings from qualitative which offers an impression that the qualitative part of this study goes beyond informing the selection of items as stated in the method section. This needs to be corrected.

Result: While I believe result section could be improved with clarity on some the comments above, I would love to know more about what Table 3 in the result section intends to impart? Why multiple responses? what is the difference between stage at which community engagement was utilized and community engagement utilized in SI? What does assess community engagement in social innovations mean? Check number of participants 369 against 274???

The qualitative findings need to align with the theme of the finding which should be aligned with the overarching questions (hopefully this could be clarified by further specifying the objectives.

Discussion and conclusion: It is important to identify some topical finding from this study and interpret each with available evidences or experiences. I am not sure if "Our findings show that half of the social innovation health projects had a very high-level engagement, categorized as either “shared leadership” or “collaborate”" While the qualitative findings and quotes do not suggest this. You may need to check your data. The statement in the conclusion section "...this study demonstrates moderate to high community engagement across various stages

of social innovation in health" doesn't seem have foundation. Neither measurements were clearer in the method section nor, the findings demonstrated this.

Reviewer #2: The manuscript is well organized, presented, and written in Standard English. The results of the research are well discussed in the light of the available related literature and the conclusions are supported by data drawn from mixed methods analysis. However, the title as well as the purpose of the manuscript needs to be specific in terms of geographical scope (such as LMICs as indicated in the manuscript). Moreover, the authors need to clearly show the originality of the results of their research based on exhaustive and critical review of available related literature (e.g. see the attached research article, which is missing in the list of references).

6. PLOS authors have the option to publish the peer review history of their article (what does this mean? ). If published, this will include your full peer review and any attached files.

**Do you want your identity to be public for this peer review?** For information about this choice, including consent withdrawal, please see our Privacy Policy .

Reviewer #1: **Yes:** Mirgissa Kaba

Reviewer #2: No

---

## [Editor Report · Decision Letter 1]

10 Feb 2026

Utilization of Community Engagement in Social Innovation Health Projects in Low- and Middle-Income Countries: A Global Sequential Mixed Methods Analysis

PGPH-D-25-00549R1

Dear Mr Ahumuza,

We are pleased to inform you that your manuscript 'Utilization of Community Engagement in Social Innovation Health Projects in Low- and Middle-Income Countries: A Global Sequential Mixed Methods Analysis' has been provisionally accepted for publication in PLOS Global Public Health.

Best regards,

Damen Haile Mariam, MD, MPH, PhD

Academic Editor